# Size-dependent stability of ultra-small α-/β-phase tin nanocrystals synthesized by microplasma

Atta Ul Haq [1], Sadegh Askari[2], Anna McLister[1], Sean Rawlinson[1], James Davis[1], Supriya Chakrabarti[1,3], Vladimir Svrcek[4], Paul Maguire [1], Pagona Papakonstantinou[1] & Davide Mariotti [1]

Nanocrystals sometimes adopt unusual crystal structure configurations in order to maintain structural stability with increasingly large surface-to-volume ratios. The understanding of these transformations is of great scientific interest and represents an opportunity to achieve beneficial materials properties resulting from different crystal arrangements. Here, the phase transformation from α to β phases of tin (Sn) nanocrystals is investigated in nanocrystals with diameters ranging from 6.1 to 1.6 nm. Ultra-small Sn nanocrystals are achieved through our highly non-equilibrium plasma process operated at atmospheric pressures. Larger nanocrystals adopt the $\beta$-Sn tetragonal structure, while smaller nanocrystals show stability with the $\alpha$-Sn diamond cubic structure. Synthesis at other conditions produce nanocrystals with mean diameters within the range 2–3 nm, which exhibit mixed phases. This work represents an important contribution to understand structural stability at the nanoscale and the possibility of achieving phases of relevance for many applications.

[1] Nanotechnology & Integrated Bio-Engineering Centre (NIBEC), Ulster University, Shore Road, Newtownabbey BT37 0QB, United Kingdom. [2] Institute for Experimental and Applied Physics, Christian-Albrechts-Universität zu Kiel, Leibnizstraße 17, 24118 Kiel, Germany. [3] Centre for Carbon Materials, International Advanced Research Centre for Powder Metallurgy and New Materials (ARCI), Balapur P.O., Hyderabad 500005, India. [4] National Institute of Advanced Industrial Science and Technology-AIST, Central 2, Tsukuba 305-8568, Japan. Correspondence and requests for materials should be addressed to A.U.H. (email: au.haq@ulster.ac.uk) or to D.M. (email: d.mariotti@ulster.ac.uk)

Tin (Sn) represents an exciting element, offering a versatile material platform to tune a range of properties. The key features of Sn are drastically affected by the crystal configuration; tin tetragonal phase ($\beta$-Sn) in the bulk form is known to be stable at room temperature and it transforms to a diamond cubic phase ($\alpha$-Sn) only below ~13 °C[1], accompanied by a large increase in volume[2]. In order to access the properties of $\alpha$-Sn at higher temperature, a range of strategies have been investigated. In particular, the transformation temperature between $\alpha$-Sn and $\beta$-Sn can be increased with the incorporation of impurity. For instance, a very low concentration (~0.6 wt%) of Si in Sn shifts the transformation temperature from ~13 to ~90 °C[3]. A major difference between the two phases of Sn is also observed in the electronic structure as the metallic behavior of tetragonal tin becomes semi-metallic when it transforms to $\alpha$-Sn (a zero-bandgap semi-metal)[2]. This results in drastically different opto-electronic properties.

These characteristics are again largely affected at the nanoscale as properties now depend on size and shape as much as they depend on the traditional parameters of structure and composition[4]. This may complicate the analysis and amplify the experimental parameter space but it also offers an opportunity for manipulating and tailoring phase transformations and corresponding properties. While there is clear experimental evidence that phase transformation of Sn nanocrystals (NCs) is different from bulk, the phase stability and transformation temperatures are essentially unknown for NCs of differing sizes and surface characteristics (e.g. free-standing or embedded). One of the few experimental efforts in this direction has for instance shown that $\alpha$-Sn NCs with sizes >8 nm are no longer energetically stable inside the embedded Si matrix and they transform into $\beta$-Sn NCs[4].

Energy storage applications have provided most of the thrust in studying Sn NCs, where alloying (e.g. with Li or Mg[4–13]) has been a key aspect to the research. Sn NCs are a promising candidate as an anode material for next generation batteries due to their high theoretical capacity (991 mAhg$^{-1}$) and low toxicity[14–16]. Also, Sn is more sustainable and environmentally friendly than carbon for Li-ion, Mg-ion and for the emerging Na-ion batteries[11,17,18]. With a theoretical capacity three times higher than graphite[16], Sn has also been used along with other technological important materials for enhancing their super-capacitive performance[13,19,20].

Another major aspect that may result from nanoscale effects is quantum confinement[21–24]. Diamond cubic Sn ($\alpha$-Sn) is a semi-metal with a low electron effective mass and excitonic confinement effects should be manifested even for relatively large NCs[25]; quantum confinement is therefore expected as the NCs size becomes much lower than the Bohr exciton radius, which is estimated to be around 12.56 nm (Supplementary Table 1). Such large value for the Bohr radius (e.g. compared to 4.5 nm for Si) makes $\alpha$-Sn more sensitive to quantum confinement effects and any phase transformation would result in a step-change for properties deriving from the electronic structure. A first outcome of quantum confinement is the widening of the bandgap and the possibility of tuning its value from 0 up to 1.24 eV[26] or more by varying the NC size. Furthermore, the properties of Sn are particularly conducive to enhancing carrier multiplication, a phenomenon that is generally negligible in bulk materials and relatively limited in NCs with various compositions (e.g. Si, PbS, PbSe)[26]. Opto-electronic properties in quantum confinement regime are particularly attractive for various applications[5,9,27–30]. For instance, $\alpha$-Sn NCs would enable the achievement of a group-IV direct bandgap semiconductor with a high degree of tunability for solar cell applications or for monolithic integration of optically active materials with silicon[25,26].

From a fundamental point of view, there has been consistent interest in Sn NCs as demonstrated by a large number of theoretical studies (e.g., ref. [2,11,26,29,31]). The experimental efforts are however limited to relatively large NCs (>17 nm)[10] or to Sn NCs embedded within bulk matrix (e.g., ref. [4,25,32,33]) or stabilized by using chelating agents[34] and dendrimer-encapsulation methods[35]. As a consequence, many of the simulation results remain unverified as these refer to unconstrained or free-standing NCs with diameters below 5 nm, where free-standing refers to NCs that are not embedded within a larger solid matrix. For instance, Hörmann et al. used density functional theory and argued the stability of only $\beta$-Sn NCs below 8 nm[31]; however, these results were in contradiction to previous experimental findings[4,10].

In our work, we study the synthesis and properties of Sn NCs with diameters from ~6.1 nm down to 1.6 nm. We therefore report the synthesis of ultra-small (<3 nm diameter) surfactant-free and free-standing Sn NCs, which is unprecedented to the best of our knowledge. We also demonstrate that $\alpha$-Sn can be stabilized at room temperature in the form of free-standing NCs with no need of a supporting matrix, which is also unprecedented to the best of our knowledge. For this purpose, we use a microplasma synthesis process that has demonstrated the required versatility to produce a range of ultra-small nanoparticles or NCs[36–49]. Microplasmas allow for the use of a wide range of precursors that can be supplied as gases, liquids or solids leading to an impressive compositional and structural NC tunability. Furthermore, we clarify the impact of quantum confinement, through bandgap widening, and nanoscale effects on the phase stability of Sn NCs and we show that room temperature stability of $\beta$-Sn is confirmed for NCs with diameter larger than ~3 nm. We note that at some of the synthesis conditions, Sn NCs with mixed phases are produced, which exhibit mean diameters in the range 2–3 nm. We finally show that the Sn NCs with either $\alpha$-Sn or $\beta$-Sn phase have exceptional electrochemical capacity, which is beneficial for supercapacitors in micro-power systems, i.e., micro-supercapacitors[50–55].

## Results

**Microplasma synthesis and characterization of tin nanocrystals.** In order to achieve accurate control in the synthesis of Sn NCs, we have used a low-cost process based on an atmospheric pressure microplasma set-up (Fig. 1). Atmospheric-pressure microplasmas offer a promising platform for the synthesis of nanomaterials due to their versatile features[36,37]. Such microplasmas at atmospheric pressures are highly non-equilibrium in nature[36,42], which are beneficial for synthesizing highly challenging nanomaterials contrary to conventional synthesis approaches. For instance, the synthesis of silicon and silicon-carbide NCs with dimensions less than 2.5 nm have been successfully presented in our recent papers[43,44]. Furthermore, the synthesis of a variety of NCs under different configurations of microplasmas have also been demonstrated along with tuning their sizes, structure and surface functionalities[43,45–47].

Our plasma reactor consists of a quartz capillary with two ring electrodes held fixed by a Perspex frame; a tin wire, used as solid precursor, is inserted in the capillary (Fig. 1a). One of the two electrodes is grounded while the other is powered with radio frequency (RF) to sustain the plasma (Fig. 1b). The inset image in Fig. 1b shows a magnified photograph where the Sn wire is clearly visible following a couple of hours of operation. Helium gas is used for the synthesis and Sn NCs were then collected directly in ethanol or directly deposited on carbon electrodes.

Sn NCs were synthesized at different He gas flow rates (0.25 sLm, 0.5 sLm, 0.75 sLm and 1 sLm) and collected directly in ethanol; the corresponding transmission electron microscopy

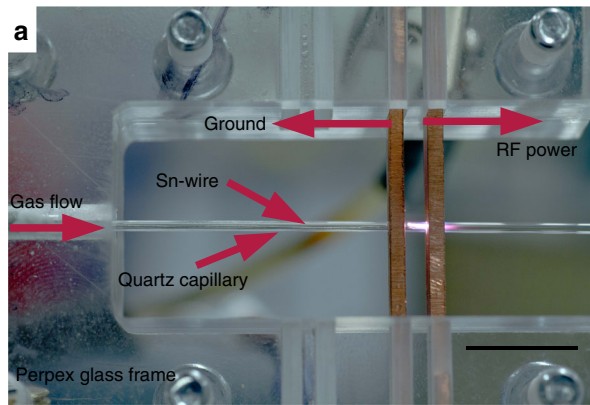
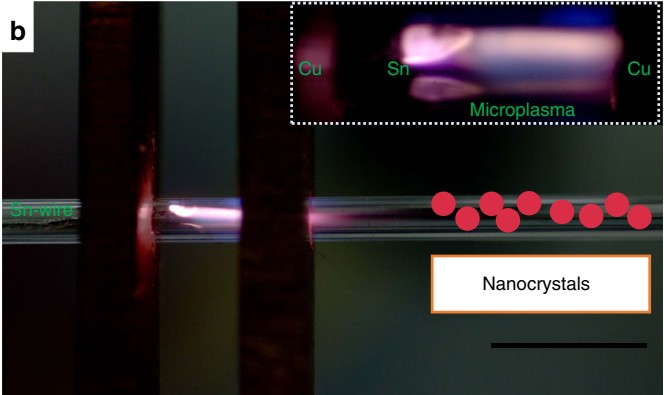

**Fig. 1** Microplasma reactor configuration at atmospheric pressure. **a** A photographic illustration of the plasma reactor set up. A tin wire (0.25 mm diameter) is placed inside the capillary such that one end of the wire is connected to ground and the other end is suspended at the middle position between the two copper electrodes; **b** A high-resolution photograph of the plasma region where the Sn-wire is clearly seen (also shown in the inset image). The scale bars in (**a**) and (**b**) are 5 and 4 mm respectively

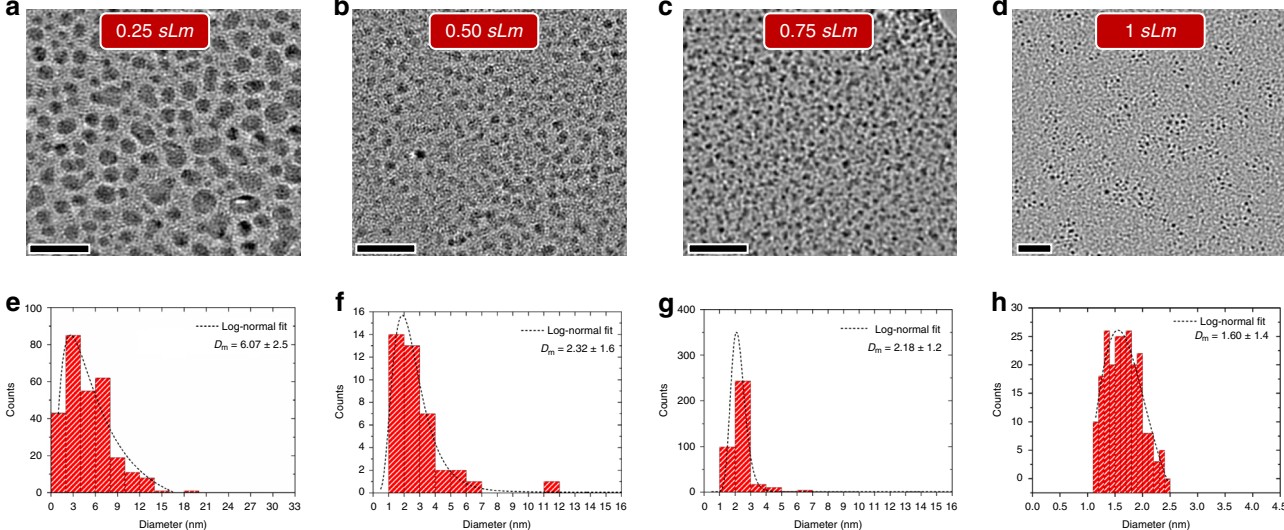

**Fig. 2** Morphology of tin nanocrystals. **a–d** Transmission electron micrographs of Sn nanocrystals (NCs) prepared at 0.25 sLm, 0.5 sLm, 0.75 sLm and 1.0 sLm respectively; (**e–h**) bar charts showing size distribution of the NCs corresponding to the same micrographs in (**a–d**) with the average size of 6.07, 2.32, 2.18 and 1.60 nm respectively. The dashed curve represents the log-normal fit to the distribution. The scale bars in (**a–d**) is equal to 20 nm and the sLm corresponds to standard liter per minute

(TEM) micrographs are shown in Fig. 2a–d. The NCs were found to be well separated for each of the gas flow conditions in the plasma. The diameter of the NCs decreases with increasing gas flow rate due to the lower residence time of the particles in the plasma (see discussion on the formation mechanisms further below). The shape of the NCs has also been affected by the gas flow rates. The NCs prepared at higher gas flow rates (e.g., 1 sLm) showed highly spherical shapes (e.g., Fig. 2d) while deviation from their spherical nature was observed at lower gas flow rates (e.g., Fig. 2a). The NCs at 0.25 sLm, where plasma heating effect is higher, were found to be slightly elongated in their shape as can be seen in Fig. 2a.

Figure 2e–h shows bar chart diagram representing the size distributions in each of the gas flow conditions. The average size of the NCs at 0.25 sLm, 0.5 sLm, 0.75 sLm and 1.0 sLm was determined by fitting log-normal distributions, which produced mean values of 6.07, 2.32, 2.18 and 1.60 nm, respectively. The size of Sn NCs increases by decreasing the flow of helium gas as shown in Fig. 2 (also summarized in Supplementary Fig. 1).

Figure 3 shows representative TEM images and selected area electron diffraction (SAED) patterns that support the existence of different phases for different synthesis conditions (see Supplementary Table 2, additional images and corresponding fast Fourier transforms in Supplementary Figs 2–9). For NCs with average diameter of 6.07 nm, the lattice fringes measured 0.20 and 0.16 nm, which correspond to the (211) and (301) planes of $\beta$-Sn, respectively (Fig. 3a). The tetragonal structure was further confirmed by SAED patterns matching with the $\beta$-Sn (JCPDS: 01-089-4898) as shown in Fig. 3e. A mixture of phases of $\alpha$-Sn and $\beta$-Sn was found for the NCs with average diameters of 2.32 and 2.18 nm (Fig. 3b, c). The lattice fringes of NCs with 2.32 nm average diameter were 0.18 and 0.29 nm that represents (222) and (200) planes of $\alpha$-Sn and $\beta$-Sn respectively as shown in Fig. 3b. The corresponding diffraction rings also matched with the crystallographic data for Sn (JCPDS/ICSD taken from Diffrac$^{plus}$ PDFMAINT: $\beta$-Sn = 01-089-4898/ 076149; $\alpha$-Sn = 01-089-4789/076040) which can be seen in Fig. 3f. Similarly, for NCs with average diameter of 2.18 nm,

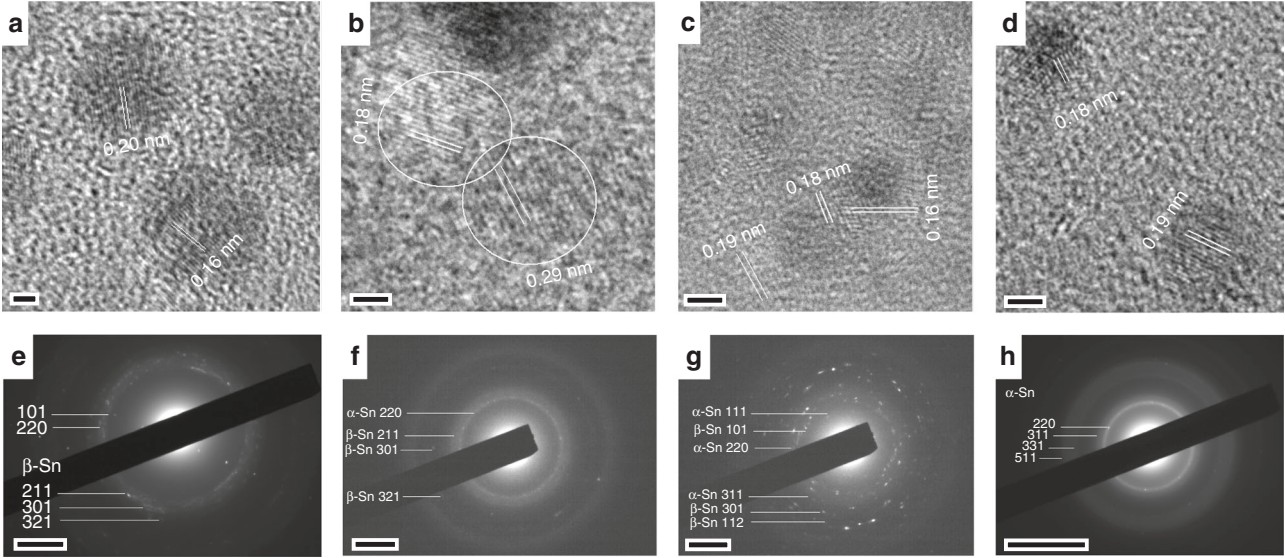

**Fig. 3** Crystal structural transformations in tin nanocrystals. **a–d** Representative high-resolution transmission electron micrographs corresponding to Sn nanocrystals (NCs) prepared at 0.25 sLm, 0.5 sLm, 0.75 sLm and 1.0 sLm and with corresponding average diameter of 6.07, 2.32, 2.18 and 1.60 nm; (**e–h**) representative selected area electron diffraction patterns of the Sn NCs. The scale bars in (**a–d**) is 2 nm, in (**e–g**) is 2 nm$^{-1}$ and in (**h**) is 5 nm$^{-1}$

lattice fringes corresponding to the planes in both $\alpha$-Sn and $\beta$-Sn were observed, i.e. 0.18 and 0.19 nm corresponding to (222) and (311) planes in $\alpha$-Sn and 0.16 nm from the (301) plane in $\beta$-Sn (Fig. 3c). The respective diffraction pattern is also shown in Fig. 3g that confirms the rings with randomly oriented crystals of mixed phases of Sn. Fig. 3d, h confirms the stability of $\alpha$-Sn at room temperature for the smallest NCs (~1.60 nm) as the lattice fringes and diffraction rings matches with the standard data charts. Lattice fringes corresponding to the planes in $\alpha$-Sn were observed, i.e. 0.18 and 0.19 nm corresponding to (222) and (311) planes (Fig. 3d). The respective diffraction pattern is also shown in Fig. 3h that confirms the rings belonging to $\alpha$-Sn.

Therefore, with the NCs diameter reduction from 6.07 to 1.60 nm, different phases were observed, i.e. $\beta$-Sn for the larger NCs, mix phases for intermediate sizes and $\alpha$-Sn for the smallest NCs. The different phases were confirmed by both the measurement of lattice fringes applied to a number of single NCs as well as by analyzing SAED patterns which depict the crystal structure of an ensemble of NCs.

In the microplasma synthesis process, Sn atoms are produced from the heated wire[56,57], which are subsequently ionized. Neutral and ionized Sn-atoms contribute to the initial phases of nucleation and then to the NCs growth. The gas flow has a direct impact on the overall background gas temperature (higher temperature for lower flow rates) where Sn-atom production rates at the wire are higher for higher temperatures. Therefore, the higher density of Sn atoms/ions produced at the wire together with a longer residence time are the reasons for larger NCs at lower gas flow rates. The opposite occurs for higher gas flow rates, with a lower Sn atoms/ions density and shorter residence time leading to smaller NCs.

As demonstrated in the TEM analysis, the larger NCs are preferably stabilized with the tetragonal phase while the diamond cubic phase is preferred at smaller sizes. It has been reported that the stabilization of thermodynamically unstable structures can be achieved by embedding them in matrix, by growing them on lattice-matched substrates and by doping or growing a shell around them[4,9,10,25,33]. In our process, at higher gas flow rates, nucleation and growth is fast whereby energy is continuously supplied to the growing NCs through frequent collisions in the plasma with a range of energetic species; this is then followed by fast quenching as the NCs are collected outside the plasma. These are common phenomena in plasma processes[58,59] which contribute to maintaining the diamond cubic structure ($\alpha$-Sn) and the crystalline nature; the conditions of these plasma processes also depart from the conditions applied in simulations which predict $\beta$-Sn NCs to be always more stable[2]. Theoretical simulations have shown that NCs with diameters in the range 1–2 nm generally tend to relax in an amorphous phase[2], while our 1.6 nm NCs confirm the contribution of plasma-induced NC selective heating and crystallization[46]. Lower gas flow rates, when higher background temperatures are expected, allow for relaxation, leading to the stabilization in the tetragonal structure. Therefore the microplasma process offers a range of unique features that have facilitated the synthesis of these NCs and which are not available to conventional techniques. The precursor supply at high pressure allows for fast aggregation kinetics instead of slow surface growth, the former resulting in ultra-small NCs while the latter generally leads to larger NCs. Selective NC heating and charging are plasma-specific effects and in this case have allowed NCs to reach the necessary crystallization temperature while remaining separate from each other. Fast quenching is another unique feature of microplasmas, which has contributed to the instantaneous stabilization of the respective NCs phases.

The mixed phases observed for the NCs with diameters in the range of 2–3 nm indicate a transition region in the synthesis parameter space. This could originate from NCs being subjected to different environments due to the strong gradients (temperatures, species densities etc.) present within the plasma volume or due to small process conditions drifts over time. The observation of mixed phases only at given synthesis conditions may suggest that some set-up improvements could extend the synthetic capabilities; at the same time it also highlights the sophistication of this process, which has shown the possibility of stabilizing of $\alpha$-/$\beta$-Sn separately at other synthesis conditions. The exploration of a larger parameter space and possibly by de-coupling the precursor supply from the nucleation/growth steps may allow extending the diameter range where either $\alpha$- or $\beta$-Sn can be stabilized. The use of liquid/gaseous precursors and the growth of

stabilizing shells are also avenues that could be investigated further.

**Quantum confinement in tin nanocrystals.** The $\alpha$-phase stabilized in our NCs (1.6 nm mean diameter) is expected to exhibit direct semiconducting behavior (vs. metallic of the $\beta$-phase) with strong quantum confinement (diameter <12.56 nm Bohr radius). We have therefore carried out measurements to determine the bandgap of the smallest $\alpha$-phase NCs and to verify the opening of the bandgap as expected from quantum confinement. Figure 4a shows the bandgap determined experimentally for the first time for the $\alpha$-Sn NCs (1.6 nm) from the UV-Vis transmission measurements and with an integrating sphere (see also Supplementary note 1 and Supplementary Fig. 10). A bandgap of 2.2 eV was estimated from the Tauc plot (Fig. 4a). The role of strong quantum confinement in our ultra-small semiconducting $\alpha$-Sn NCs opens the bandgap as expected also from theoretical calculations reported by Allan et al.[26]. In Fig. 4b we report the bandgap values reported by Allan et al.[26] (highlighted inside the blue rectangular region), which were calculated for diameters >2 nm. In order to compare with our smaller NCs we have extrapolated

to 1.6 nm by fitting with an exponential curve. This shows that our experimental results are in very close agreement with the theoretical calculations reported by Allan et al.[26].

**Electrochemical performance of tin nanocrystals.** Owing to the energetic and high surface area of our plasma-produced Sn NCs and their practical importance in many energy-related devices, we finally assess the properties of our Sn NCs for one of their potential applications. In particular, we tested the electrochemical capacitance of Sn NCs (see Supplementary Figs 11–13), which is relevant for application as an electrode material for micro-supercapacitors[60,61]. Fig. 5a shows the cyclic voltammograms of 1.6 and 6.1 nm Sn NCs at a scan rate of 40 mV s$^{-1}$. The potential window range was taken from −0.2 to 0.1 V. It can be noted that the shape of the curves show a rectangular capacitive behavior depicting fast and easy diffusion/migration of charges[62]; this is representative of a highly reversible non-faradaic reaction at the electrode/electrolyte interface, which is characteristic of electrical double layer capacitors[61].

The areal capacitance of these NCs was estimated (see Supplementary note 2) with the active electrode area of

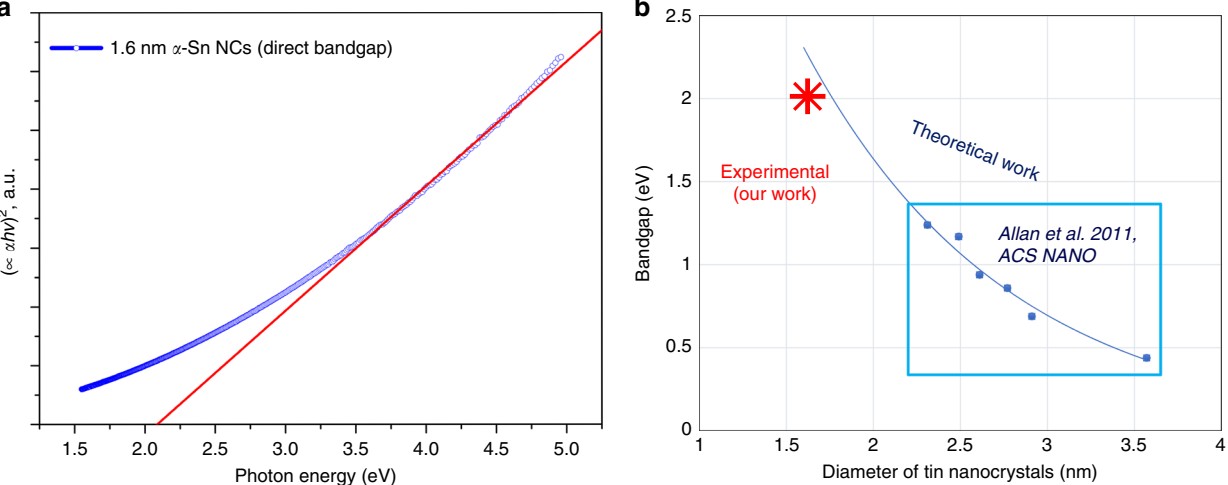

**Fig. 4** Quantum confinement induced bandgap widening in tin (Sn) nanocrystals (NCs). **a** Tauc plot of 1.6 nm Sn NCs and linear fit which estimates the bandgap to be 2.2 eV; as per Tauc plot of direct semiconductors, $C'$ is the proportionality constant, $\alpha$ is the absorption coefficient, $h\nu$ is the energy ($h$ is Planck's constant and $\nu$ is the frequency). **b** Bandgap values (blue dots) for Sn NCs from theoretical calculations[26] and fitted exponential curve (blue line). The red star corresponds to our experimental result

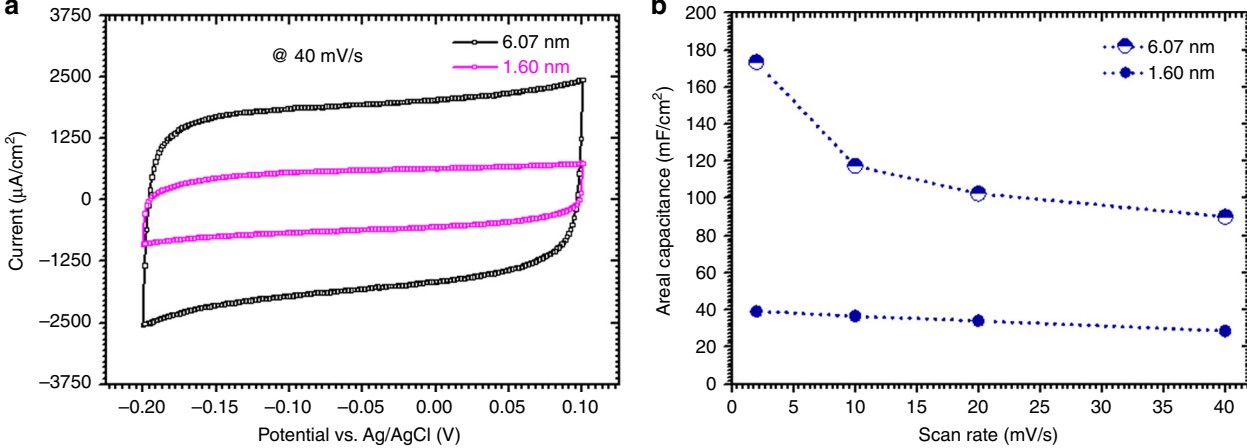

**Fig. 5** Electrochemical performance of tin (Sn) nanocrystals. **a** Cyclic voltammograms of tin (Sn) nanocrystals (NCs) in 0.1 M sodium hydroxide (NaOH) solution attained at scan rate 40 mV s$^{-1}$ respectively; **b** evolution of the relative areal capacitances of Sn NCs versus scan rates

~0.04 cm$^2$; the areal capacitance is a more significant performance metric for micro-supercapacitors (e.g. vs. gravimetric-based measurements) as the weight of the active material is negligible and the available area is generally limited for integrating components into a system[50,51,53,55,63]. As can be seen in Fig. 5b, a remarkably high capacitance of 173 mF cm$^{-2}$ and 39 mF cm$^{-2}$ was measured, at nearly equilibrium conditions (~2 mV s$^{-1}$); this is the highest areal capacitance measured for any Sn-based electrode, to the best of our knowledge. Furthermore, our reported areal capacitance values are also much higher than the carbon electrodes (see Supplementary Fig. 14) and those of the reported values for metallic nanoparticle-based, carbon-based supercapacitors and others[50,53–55,63–67]. Although, ruthenium oxide (RuO$_2$) has been reported[53,68,69] with higher capacitance, Sn-based devices still present much greater benefits originating from the much lower cost and much higher availability of Sn vs. Ru. It should also be noted that RuO$_2$ devices rely on reversible faradaic surface reactions (i.e. pseudo-capacitors) and therefore their scan rates are intrinsically limited[53] compared to those available with electrical double-layers.

The higher capacitance for larger NCs (6.07 nm) compared to that from smaller NCs (1.60 nm) could be due to the relatively lower intrinsic electrical resistivity of purely metallic ($\beta$-Sn) compared to that of the semiconducting phase ($\alpha$-Sn) that may affect the charge transport to the current collector[10]. The smaller NCs with a diamond cubic phase ($\alpha$-Sn) show overall excellent capacitive performance with the capacitance values still exceptionally high and nearly constant with increasing scan rates (Fig. 5b). The capacity retention in the larger $\beta$-Sn NCs is decreasing with increasing scan rate (2–40 mV s$^{-1}$); this is related to insufficient time for the electrolyte to adsorb and desorb on the electrode surface and an increased resistance. This increase in the internal resistance is expected due to the poor packing of the larger NCs along with their lower surface area compared to smaller NCs as reported elsewhere[70].

## Discussion

In this work, size and structure selective NCs of Sn have been successfully synthesized using plasma synthesis operated at atmospheric pressure. The size and crystalline phase of the NCs was strongly dependent on the gas flow rate. NCs produced with flow rates at 0.25 sLm, 0.5 sLm, 0.75 sLm and 1.0 sLm resulted in NCs with mean diameters of 6.07, 2.32, 2.18 and 1.60 nm, respectively. The lowest flow rate, produced NCs of the largest mean diameter and of $\beta$-Sn phase; increasing the flow rate resulted in NCs with progressively smaller diameters (2–3 nm) and mixed $\alpha$-/$\beta$-Sn phases. Finally, the higher gas flow rate resulted in much smaller NCs with regular and spherical shapes and with a diamond cubic structure of Sn. The stabilization of $\alpha$-Sn was found to be at 1.60 nm. A phase transition was observed from diamond cubic ($\alpha$-Sn) to tetragonal structure ($\beta$-Sn) with decreasing gas flow rates. The experimental results first demonstrate the possibility of stabilizing $\alpha$-Sn in free-standing and surfactant-free NCs without any supporting matrix. The smallest NCs (1.6 nm, $\alpha$-Sn) also exhibit quantum confinement and semiconducting behavior with a bandgap widening as predicted by theoretical calculations. Areal capacitances of 173 mF cm$^{-2}$ and 39.2 mF cm$^{-2}$ were estimated for larger (6.07 nm) and smaller NCs (1.60 nm) respectively. While further work and full device characterization will be required, the high areal capacitance of 173 mF cm$^{-2}$ represents the best value achieved for Sn NC electrodes, to the best of our knowledge, and compares favorably with other metallic nanoparticle and carbon-based supercapacitor electrodes. The smaller NCs also exhibited

retention of the areal capacitance for a wide range of scan rates. The results also show that plasma-based processes allow for exquisite synthesis capabilities leading to Sn NCs with unprecedented dimensions; in addition atmospheric pressure operation and the use of a solid precursor offer clear advantages for the deployment and direct integration in device manufacturing[52]. The work therefore advances the current state of the art, provides opportunities to compare and verify theoretical results and offer important directions for applications.

## Methods

**Details of the microplasma synthesis process.** Our plasma reactor consists of a quartz capillary (100 mm long) with an internal and external diameter of 0.75 and 1.0 mm, respectively (Fig. 1). Two ring electrodes with the thickness of 1 mm each and with a spacing of 2 mm are held fixed around the capillary by a Perspex frame. A tin (Sn) wire (0.25 mm diameter, Goodfellow Ltd.) is inserted in the capillary such that one end is kept fixed at 1 mm from the powered electrode. The Sn-wire acts as a sacrificial electrode and Sn precursor. A radio frequency (RF) power of 40 W (applied power) was selected as an optimized power for sustaining the plasma as shown in Fig. 1b. The flow rate of helium gas was varied from 0.25 sLm to 1.00 sLm. The plasma process produces 2–6 μg min$^{-1}$ depending on synthesis conditions (Supplementary note 3 and Supplementary Table 3). This process has been optimized for investigating the fundamental nature of these small NCs. However scaling the process throughput will be possible through known approaches involving e.g. plasma geometry expansion in one dimension[39,71] or by using multiple parallel plasma reactors arrays[36,48,72]. In any such development, the power delivery and flow will require enhanced feedback control to maintain the plasma conditions within the precise parameter range discovered here.

**Materials Characterization.** The morphology and structure of the Sn NCs were examined by TEM and SAED using a JEOL JEM-2100F electron microscope operated at 200 keV. The samples for TEM were prepared by drop casting Sn NCs/ethanol colloids on holy carbon mesh/Cu TEM grids and then were allowed to dry. The analysis of the size distribution of the Sn NCs was carried out by processing TEM micrographs with ImageJ software. The lattice fringes were measured using Gatan Microscopy Suit Software while the analysis of SAED patterns were analyzed manually by measuring the diameters of each ring and comparing them with standard crystallographic JCPDS cards.

Ultraviolet-visible (UV-Vis) spectroscopy measurements were carried out on a PerkinElmer Lambda 650S accessorized with 150 mm integrating sphere. Sn NCs were deposited directly on quartz slides to form a film. The measurements were carried out in standard transmission mode first and then within the integrating sphere to account for reflection (see Supplementary note 4 for more details). The electrochemical characterization was performed in 0.1 M sodium hydroxide using cyclic voltammetry with a conventional three electrode cell in which platinum wire and Ag/AgCl (3 M NaCl) half-cell served as counter and reference electrode respectively (Supplementary Fig. 11). All the electrochemical measurements were made using a Type III Micro Autolab (EcoChimie, Netherlands) potentiostat. The samples for cyclic voltammetry (CV) analysis were prepared by directly depositing Sn NCs on a 0.04 cm$^2$ area of screen printed carbon electrodes.

## Data availability

The data that support the findings of this study are available within the article and the Supplementary Information, and available from the corresponding author upon request

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

## Acknowledgements

This work was supported by the Marie Curie Initial Training Network (RAPID-ITN, award n.606889) and by EPSRC (awards n.EP/K022237/1, n.EP/M024938/1 and n. EP/M015211/1). We would like to acknowledge Beatrix Biskup (Ruhr-university Bochum, Germany) for taking the photos in Fig. 1.

## Author contribution

A.H. designed, performed experiments, collected data, analyzed and wrote the manuscript; S.A. contributed with TEM analysis; A.M., S.R. & J.D. contributed with the electrochemical measurements; S.C. & P.P. contributed to the analysis of the electrochemical results; D.M. V.S. & P.M. contributed to the initial research idea; D.M. supervised the project and co-wrote/reviewed the paper. All authors contributed to the write-up and provided feedback to the final version of the manuscript.

## Additional information

**Competing interests:** The authors declare no competing interests

