## [Peer Review File · Nature Communications]

Reviewers' Comments:

Reviewer #1:

Remarks to the Author:

This work represents an advancement in the field of synthesis of Sn nanoparticles, and merits publication in Nature Comm.

The authors have done a good job in their literature review and discuss the challenges of synthesizing alpha-Sn in particular (as opposed to beta-Sn). Before this work, alpha-Sn could not be stabilized at room temperature without being confined to a matrix. They have managed to synthesize freestanding alpha-Sn at room temperature which is very exciting for batteries and many other applications.

While the authors state that their larger particles are beta-Sn and their smallest (1.6nm) particles are alpha-Sn, they don't emphasize in their introduction or conclusions that the sizes in between have a mixture of phases, although they discuss this clearly within the manuscript. I think it is important to point this out as an additional challenge of the synthesis process of these materials, and should be made clear in the abstract and conclusions, with perhaps a discussion on whether there is a way to remove the mixed phases and make pure alpha-Sn with variable (smaller) size nanoparticles in the future. This would be very beneficial for applications.

Lastly, although the analysis of the diffraction patterns appears to be thorough, it is difficult for me to verify the fringes in the TEM images. I also cannot verify the shapes, which is another indicator of alpha vs. beta Sn. I would like to see higher (pixel) resolution images, higher magnification images, and/or STEM images of the particles in the manuscript so that readers can more easily verify the structure for themselves visually.

With those points addressed, I would be pleased to see this work accepted into Nature Comm.

Reviewer #2:

Remarks to the Author:

The paper reports very interesting results on achieving a phase transition / selectivity by quantum effects, i.e. bringing a material down to nanometer size, which is quite relevant for a number of practical applications. While a myriad of papers is available on Ag and Au nanoparticles, Sn nanoparticles have not so much been investigated in the past, which contributes to the uniqueness to the paper. The authors use a new approach for nanoscale manufacture which is a microplasma. This adds to the innovation.

The paper is well readable and gives a consistent and complete innovation story. I feel the authors meet the high targets of the journal. I thus recommend publication.

- Why not mentioning the use of microplasma in the title? This is an essential approach.
- I miss the justification why microplasma is needed and not a conventional, e.g. wet-chemical, method to produce the nanoparticles.
- This would allow a clear benchmarking of plasma versus conventional technology for Sn

particles

- How can such process scaled-up? This should be given as outlook. The semiconductor market with large volumes is targeted.
- The authors should explain what "free standing" means.
- The authors talk a lot about quantum effects. Basically they show often that when particles approaches nanoscale all is different to what is known. It would be appreciated if they could provide a (theoretical) explanation for this. Some review papers deal with such information.
- I searched for quite a while from which precursor the Sn nanoparticles are made. It might be mentioned. Yet it is hidden. Give it where it should be = in the Experimental section.
- Highly environmental process = be careful with such claims. Is there any LCA study which proves that? Then discuss and cite it. Otherwise say for this and that indication and guesstimate, there is a likelihood for ...
- Nicolas G. = improper way of citation. Only surnames have to be given.
- I feel more microplasma work should be cited and discussed in the introduction and it should be made clear what this technology can do.

Reviewer #3:

Remarks to the Author:

This manuscript described the synthesis of Sn nanocrystals using micro-plasma reactor and showed micro-supercapacitor application. The nanocrystals are characterized using TEM and UV, and the electrochemical performances are investigated using CV technique.

1. The nanocrystals was synthesized at different He gas flow. Is there any effect of gas (e.g, Ar or N₂) or flow rate? Could you possibly make the larger size of Sn than 6.07 nm?
 2. Did the authors check the production rate?
 3. The authors compared supercapacitor performance of Sn with different sizes. However, it is difficult to compare areal capacitances of different-sized Sn samples. When preparing the electrodes, the loading mass of electrode materials may be different.
 4. The present data for supercapacitor application is not sufficient to demonstrate the potential of Sn nanocrystals as electrode materials. The rate performance (Figure 5b) is not good, and the long-term stability is not demonstrated in this manuscript.
- Therefore, I think it was premature to publish the current state of manuscript.

We would like to thank the editor and reviewers for taking their time and providing us with their feedback. We have looked carefully at all the comments and suggestions and we have produced a revised version of our manuscript where we have introduced corresponding corrections and improvements. Here below we provide replies and describe the changes we introduced in the manuscript.

Reviewer #1

This work represents an advancement in the field of synthesis of Sn nanoparticles, and merits publication in Nature Comm. The authors have done a good job in their literature review and discuss the challenges of synthesizing alpha-Sn in particular (as opposed to beta-Sn). Before this work, alpha-Sn could not be stabilized at room temperature without being confined to a matrix. They have managed to synthesize freestanding alpha-Sn at room temperature which is very exciting for batteries and many other applications. While the authors state that their larger particles are beta-Sn and their smallest (1.6nm) particles are alpha-Sn, they don't emphasize in their introduction or conclusions that the sizes in between have a mixture of phases, although they discuss this clearly within the manuscript. I think it is important to point this out as an additional challenge of the synthesis process of these materials, and should be made clear in the abstract and conclusions, with perhaps a discussion on whether there is a way to remove the mixed phases and make pure alpha-Sn with variable (smaller) size nanoparticles in the future. This would be very beneficial for applications.

Our Reply

We thank the reviewer for acknowledging the quality of our work and for highlighting the suitability for publication in Nature Communications. We have now emphasized the presence of mixed phases at some of the synthesis conditions in the abstract, introduction and conclusion. The following text has been now included/modified, respectively:

“Synthesis at other conditions produced NCs with mean diameters within the range 2-3 nm, which exhibited mixed phases.” (in the abstract).

“We note that at some of the synthesis conditions, Sn NCs with mixed phases were produced, which exhibited mean diameters in the range 2-3 nm. We finally show that the Sn NCs with either α -Sn or β -Sn phase have exceptional electrochemical capacity, which is beneficial for supercapacitors in micro-power systems, i.e. micro-supercapacitors⁵⁰⁻⁵⁵.” (in the introduction).

“The lowest flow rate, produced NCs of the largest mean diameter and of β -Sn phase; increasing the flow rate resulted in NCs with progressively smaller diameters (2-3 nm) and mixed α -/ β -Sn phases. Finally, the higher gas flow rate resulted in much smaller NCs with regular and spherical shapes and with a diamond cubic structure of Sn.” (in the conclusion).

We have also included a discussion on the possibility of producing alpha-Sn with variable sizes before figure 4 where the following has been added:

“The mixed phases observed for the NCs with diameters in the range of 2-3 nm indicate a transition region in the synthesis parameter space. This could originate from NCs being subjected to different environments due to the strong gradients (temperatures, species densities etc.) present within the plasma volume or due to small process conditions drifts over time. The observation of mixed phases only at given synthesis conditions may suggest that some set-up improvements could extend the synthetic capabilities; at the same time it also highlights the sophistication of this process, which has shown the possibility of stabilizing of α -/ β -Sn separately at other synthesis conditions. The exploration of a larger parameter space and possibly by de-coupling the precursor supply from the

nucleation/growth steps may allow extending the diameter range where either α - or β -Sn can be stabilized. The use of liquid/gaseous precursors and the growth of stabilizing shells are also avenues that could be investigated further.”

Lastly, although the analysis of the diffraction patterns appears to be thorough, it is difficult for me to verify the fringes in the TEM images. I also cannot verify the shapes, which is another indicator of alpha vs. beta Sn. I would like to see higher (pixel) resolution images, higher magnification images, and/or STEM images of the particles in the manuscript so that readers can more easily verify the structure for themselves visually. With those points addressed, I would be pleased to see this work accepted into Nature Comm.

Our Reply

We have now included a range of higher resolution and higher magnification images as well as FFTs at the end of the supplementary information; these should allow verifying fringes and structure.

Reviewer #2

The paper reports very interesting results on achieving a phase transition / selectivity by quantum effects, i.e. bringing a material down to nanometer size, which is quite relevant for a number of practical applications. While a myriad of papers is available on Ag and Au nanoparticles, Sn nanoparticles have not so much been investigated in the past, which contributes to the uniqueness to the paper. The authors use a new approach for nanoscale manufacture which is a microplasma. This adds to the innovation. The paper is well readable and gives a consistent and complete innovation story. I feel the authors meet the high targets of the journal. I thus recommend publication. - Why not mentioning the use of microplasma in the title? This is an essential approach.

Our Reply

We thank the reviewer for considering our work very interesting and for recommending our manuscript for publication in Nature Communications. We have now included the use of microplasma in the title.

- I miss the justification why microplasma is needed and not a conventional, e.g. wet-chemical, method to produce the nanoparticles.

- This would allow a clear benchmarking of plasma versus conventional technology for Sn particles

Our Reply

We have included more details on this aspect and the following text has been included (see at page 11):

“Therefore the microplasma process offers a range of unique features that have facilitated the synthesis of these NCs and which are not available to conventional techniques. The precursor supply at high pressure allows for fast aggregation kinetics instead of slow surface growth, the former resulting in ultra-small NCs while the latter generally leads to larger NCs. Selective NC heating and charging are plasma-specific effects and in this case have allowed NCs to reach the necessary crystallization temperature while remaining separate from each other. Fast quenching is another unique feature of microplasmas, which has contributed to the instantaneous stabilization of the respective NCs phases.”

- How can such process scaled-up? This should be given as outlook. The semiconductor market with large volumes is targeted.

Our Reply

We have included more details on the possibility of scaling up, see below and at the end of section 2.1:

“The plasma process produces 2-6 $\mu\text{g min}^{-1}$ depending on synthesis conditions (see supporting information). This process has been optimized for investigating the fundamental nature of these small NCs. However scaling the process throughput will be possible through known approaches involving e.g. plasma geometry expansion in one dimension^{39,56} or by using multiple parallel plasma reactors arrays.^{36,48,57} In any such development the power delivery and flow will require enhanced feedback control to maintain the plasma conditions within the precise parameter range discovered here.”

- The authors should explain what "free standing" means.

Our Reply

This is now explained as follows at page 4:

“As a consequence, many of the simulation results remain unverified as these refer to unconstrained or free-standing NCs with diameters below 5 nm, where free-standing refers to NCs that are not embedded within a larger solid matrix.”

- *The authors talk a lot about quantum effects. Basically they show often that when particles approaches nanoscale all is different to what is known. It would be appreciated if they could provide a (theoretical) explanation for this. Some review papers deal with such information.*

Our Reply

We have now included additional references, including textbook that deal with the theoretical aspect of this subject. See references [21–24].

- *I searched for quite a while from which precursor the Sn nanoparticles are made. It might be mentioned. Yet it is hidden. Give it where it should be = in the Experimental section.*

Our Reply

A tin (Sn) wire is used as precursor. We have now emphasized the role of the tin (Sn) wire as precursor and have changed the text in the experimental section as follows:

“A tin (Sn) wire (0.25 mm diameter, Goodfellow Ltd.) is inserted in the capillary such that one end is kept fixed at 1 mm from the powered electrode. The Sn-wire acts as a sacrificial electrode and Sn precursor.”

This use of wire as the Sn precursor is mentioned both within the manuscript (e.g. *“... a tin wire, used as solid precursor, is inserted in the capillary (Figure 1a)”*), in the experimental section as well as shown/mentioned in figure 1a and 1b.

- *Highly environmental process = be careful with such claims. Is there any LCA study which proves that? Then discuss and cite it. Otherwise say for this and that indication and guesstimate, there is a likelihood for ...*

Our Reply

Thanks for pointing this out. We have rephrased our statement, which now reads:

“In order to achieve accurate control in the synthesis of Sn NCs, we have used a low-cost process based on an atmospheric pressure microplasma set-up (Figure 1).”

- *Nicolas G. = improper way of citation. Only surnames have to be given.*

Our Reply

We have now corrected this.

- *I feel more microplasma work should be cited and discussed in the introduction and it should be made clear what this technology can do.*

Our Reply

These aspects were discussed at the beginning of section 3 with relevant references [54-60] in the initial submission. We have therefore addressed this also in the introduction as the reviewer

suggested, including a few more references. The following text is now included toward the end of page 4:

“For this purpose we have used a microplasma synthesis process that has demonstrated the required versatility to produce a range of ultra-small nanoparticles or NCs.³⁶⁻⁴⁹ Microplasmas allows for the use of a wide range of precursors that can be supplied as gases, liquids or solids leading to an impressive compositional and structural NC tunability.”

Reviewer #3

This manuscript described the synthesis of Sn nanocrystals using micro-plasma reactor and showed micro-supercapacitor application. The nanocrystals are characterized using TEM and UV, and the electrochemical performances are investigated using CV technique. 1. The nanocrystals was synthesized at different He gas flow. Is there any effect of gas (e.g, Ar or N2) or flow rate? Could you possibly make the larger size of Sn than 6.07 nm?

Our Reply

The effect of different gas mixtures of course has substantial impact on the plasma conditions; while viable argon plasmas are generally achievable, the use of nitrogen is limited to low concentrations in this set-up. Nitrogen is not an inert gas and is likely to produce undesirable impurities in the context of this work. We have not studied the impact of varying gas mixtures in this paper. Since the inclusion of trace gases as mixtures can have significant effects on plasma conditions, their study would expand the research and discussion into more fundamental plasma topics, which are not the focus of the work at this stage.

The effect of flow rate is indeed reported and we have discussed in the manuscript how this parameter affects the size of the Sn NCs. The impact of this in the synthesis process is also discussed in the manuscript, for instance:

“The gas flow has a direct impact on the overall background gas temperature (higher temperature for lower flow rates) where Sn-atom production rates at the wire are higher for higher temperatures. Therefore, the higher density of Sn atoms/ions produced at the wire together with a longer residence time are the reasons for larger NCs at lower gas flow rates. The opposite occurs for higher gas flow rates, with a lower Sn atoms/ions density and shorter residence time leading to smaller NCs.”

In principle it should be possible to produce larger NCs, however the investigation of larger sizes was not the focus of this manuscript. There is a strong interest in both quantum confinement and the alpha-Sn phase and these are directly linked with smaller sizes. We have however addressed this comment and discuss aspects that relate to the synthesis of different sizes, see before figure 4 where the following has been added:

“The mixed phases observed for the NCs with diameters in the range of 2-3 nm indicate a transition region in the synthesis parameter space. This could originate from NCs being subjected to different environments due to the strong gradients (temperatures, species densities etc.) present within the plasma volume or due to small process conditions drifts over time. The observation of mixed phases only at given synthesis conditions may suggest that some set-up improvements could extend the synthetic capabilities; at the same time it also highlights the sophistication of this process, which has shown the possibility of stabilizing of α -/ β -Sn separately at other synthesis conditions. The exploration of a larger parameter space and possibly by de-coupling the precursor supply from the nucleation/growth steps may allow extending the diameter range where either α - or β -Sn can be stabilized. The use of liquid/gaseous precursors and the growth of stabilizing shells are also avenues that could be investigated further.”

2. Did the authors check the production rate?

Our Reply

Yes and we have now reported this in the manuscript at the end of section 2.1:

“The plasma process produces 2-6 $\mu\text{g min}^{-1}$ depending on synthesis conditions (see supporting information).”

3. *The authors compared supercapacitor performance of Sn with different sizes. However, it is difficult to compare areal capacitances of different-sized Sn samples. When preparing the electrodes, the loading mass of electrode materials may be different.*

Our Reply

With regard to the use of areal capacitance in our results, we believe that this is the correct figure of merit as discussed by Gotsoi Y. *et al.* (Science 334, 2011, 917 [33]): "... *the gravimetric energy density is almost irrelevant compared to areal or volumetric energy for microdevices and thin film [electrochemical capacitors]...*". This is because specific energy may be of limited importance in assessing newer trends in energy storage such as for micro-supercapacitors (Science 335, 2012, 1312 [34]); the mass of the material used in these devices is so small that is negligible in the context of the application where space and specifically surface area is what matters (see also page 13 where references [35,37,46] are cited).

4. *The present data for supercapacitor application is not sufficient to demonstrate the potential of Sn nanocrystals as electrode materials. The rate performance (Figure 5b) is not good, and the long-term stability is not demonstrated in this manuscript. Therefore, I think it was premature to publish the current state of manuscript.*

Our Reply

We have reviewed our manuscript in relation to the supercapacitor application; text has been modified throughout (abstract, introduction, page 13 and conclusion) to ensure that our discussion on the potential application of our NCs for supercapacitor is valid. Furthermore, we have now included the performance of bare electrodes in the supporting information which highlights the significant improvement due to the presence of Sn NCs.

We would like to stress that our paper, as obvious from the title and abstract and throughout the emphasis in the full manuscript, is focused on the synthetic capabilities and stabilization of the alpha-phase of tin; these are our core findings. The estimation of the bandgap (figure 4) and the electrochemical capacitance (figure 5) are indeed very important results that highlight the potential applications of this new stabilized phase and reaffirm the impact of our work. We do not believe that it would be within the scope of this communication to introduce a full device characterization of the electrochemical capacitor suggested in the context of figure 5 or a full study of solar cell devices related to figure 4. In this context our results are justified and correct.

Our CV measurements are aimed at the evaluation of the electrochemical response of our NCs suggesting their possible implication as an electrode material in supercapacitors; this is customary practice in many high quality publications that focus on materials developments. We have previously carried out an extensive literature review which showed that our reported areal capacitance values are also much higher than the reported values for metallic nanoparticle-based (e.g. B, Ag and Au), carbon-based supercapacitors and others (see references [32,35–37,46–50] of our initial submission). Although, ruthenium oxide (RuO₂) has been reported with higher capacitance, substantial research is being conducted to find a replacement material at a much lower cost and with much higher availability, such as Sn. It should also be noted that RuO₂ devices rely on reversible faradaic surface reactions (i.e. pseudo-capacitors) and therefore their scan rates are intrinsically limited compared to those available with electrical double-layers.

With regard to the rate performance, capacitance generally decreases with increasing the scan rate as can be seen in most of the papers dealing with supercapacitors (ACS Appl. Mater. Interfaces 2015, 7, 14843–14850; J. Mater. Chem. A, 2013, 1, 12962–12970; AIP Conference Proceedings 1538, 228, 2013; Nature communications 4:2487, 2013, DOI: 10.1038/ncomms3487; small 2013, 9, No. 22,

3829–3833; references 55,62,63 in the main manuscript). In fact, in our case the capacitance of Sn NCs still show outstanding capacitance with increasing rates and are far better than the bare carbon electrodes as can be seen in our new results included in the supporting information.

Reviewers' Comments:

Reviewer #1:

Remarks to the Author:

I looked over the manuscript revisions and I believe they have addressed all my concerns, and it looks like they addressed the other reviewer's concerns as well. It is always helpful to me to see the other reviewer's comments as well, and how the authors responded to it. I am fine with proceeding to acceptance.

Reviewer #3:

Remarks to the Author:

This manuscript was well revised according to the reviewer's comments. Now, I would like to suggest this manuscript for publication in Nature Communications.